# PROBABILISTIC VIEW OF MULTI-AGENT REINFORCEMENT LEARNING: A UNIFIED APPROACH

## ABSTRACT

Formulating the reinforcement learning (RL) problem in the framework of probabilistic inference not only offers a new perspective about RL, but also yields practical algorithms that are more robust and easier to train. While this connection between RL and probabilistic inference has been extensively studied in the single-agent setting, it has not yet been fully understood in the multi-agent setup. In this paper, we pose the problem of multi-agent reinforcement learning as the problem of performing inference in a particular graphical model. We model the environment, as seen by each of the agents, using separate but related Markov decision processes. We derive a practical off-policy maximum-entropy actor-critic algorithm that we call *Multi-agent Soft Actor-Critic* (MA-SAC) for performing approximate inference in the proposed model using variational inference. MA-SAC can be employed in both cooperative and competitive settings. Through experiments, we demonstrate that MA-SAC outperforms a strong baseline on several multi-agent scenarios. While MA-SAC is one resultant multi-agent RL algorithm that can be derived from the proposed probabilistic framework, our work provides a unified view of maximum-entropy algorithms in the multi-agent setting.

## 1 INTRODUCTION

The traditional reinforcement learning (RL) paradigm, that formalizes learning based on trial and error, has primarily been developed for scenarios where a single trainable agent is learning in an environment. In this setup, although the agent changes its behavior as learning progresses, the environment dynamics themselves do not change. Thus, the environment appears to be *stationary* from the point of view of the learning agent. However, in a setting where several agents are learning in the same environment simultaneously (multi-agent setting), this is not true as a change in one agent's behavior manifests itself as a change in environment dynamics from the point of view of other agents (Busoniu et al., 2008). It has been established that stability issues can arise if each agent is independently trained using standard single-agent RL methods (Tan, 1993).

While, in theory, it is possible to treat a collection of multiple agents as a single centralized meta-agent to be trained, in practice, this approach becomes infeasible as the action space of the centralized meta-agent grows exponentially with the number of agents. Moreover, executing the resultant centralized policy is not always possible due to various reasons like geographic separation between agents, communication overhead and so on (Foerster et al., 2018b). Even if these issues are taken care of, when the agents are competitive, designing a reward function for the centralized meta-agent is very challenging and thus, in general, such a setup cannot be used with competitive agents.

There are numerous practical scenarios that require several intelligent agents to function together (either cooperatively or competitively). Consider, for instance, a soccer game between two teams: agents within a team must cooperate while being competitive with the opponents. Considering that traditional single-agent RL methods cannot satisfactorily handle problems from the multi-agent domain, completely new RL algorithms that explicitly acknowledge and exploit the presence of other intelligent agents in the environment are required.

In this paper, we pose the multi-agent reinforcement learning problem as the problem of performing probabilistic inference in a particular graphical model. While such a formulation is well known in the single-agent RL setting (Levine, 2018), its extension to the multi-agent setup is non-trivial especially in the general case where the agents may be cooperative and/or competitive. We model

the environment as seen by each of the agents using separate but related Markov Decision Processes (MDPs). Each agent then tries to maximize the expected *return* it gets from the environment under its own MDP (Section 4).

Using our framework, we derive an off-policy maximum entropy actor-critic algorithm that generalizes the Soft Actor-Critic (SAC) algorithm (Haarnoja et al., 2018a) to a multi-agent setup. We refer to this algorithm as *Multi-agent Soft Actor-Critic* (MA-SAC). Like SAC, it is a maximum entropy algorithm, i.e., the learned policies try to maximize the rewards while at the same time maximizing entropy of the stochastic actor. Such algorithms are known to be more stable and easier to train (Haarnoja et al., 2018a).

MA-SAC follows the *centralized training, decentralized execution* paradigm. As we demonstrate in Section 4, each agent learns its own policy while being actively aware of the presence of other agents. The learned policy of any given agent only utilizes its local observation at test time. Thus, MA-SAC avoids the pitfalls of both independent training of agents (being unaware of other agents leads to non-stationarity and hence instability) and training a centralized agent (centralized policies are hard to execute) as described above.

By setting a tunable temperature parameter (Section 4.3) to zero, MA-SAC yields an algorithm that is very similar to the Multi-agent Deep Deterministic Policy Gradients algorithm (MADDPG) (Lowe et al., 2017) apart from a minor change in updating the actor. The utility of this modification is clearly reflected in our derivation of the inference procedure. When the temperature parameter is non-zero, agents trained using MA-SAC outperform agents trained using MADDPG on multiple cooperative and competitive tasks as we demonstrate in Section 5.3.

Our main contributions are: **(i)** we present a probabilistic view of the multi-agent reinforcement learning problem where each agent models the environment using a separate but related MDP; **(ii)** we derive an off-policy maximum entropy actor-critic algorithm (MA-SAC) by performing structured variational inference in the proposed model; **(iii)** we empirically demonstrate that MA-SAC performs well in practice and highlight different ways in which our framework can utilize ideas from other existing approaches in multi-agent RL; and **(iv)** although we only present an actor-critic algorithm in this paper, our framework allows derivation of maximum-entropy variants of other reinforcement learning algorithms in the multi-agent setting.

## 2 RELATED WORK

In recent years, RL has made significant progress in complicated domains such as game playing (Mnih et al., 2015; Silver et al., 2016) and robotics (Levine et al., 2016) to name a few. Inspired by these successes, researchers have tried independently training all agents in a multi-agent setup using single-agent RL algorithms as in the case of independent $Q$-learning (Tan, 1993) and independent deep $Q$-learning (Tampuu et al., 2017). However, as discussed in Section 1, such an approach does not perform well in practice. Recently, Foerster et al. (2018a) have proposed a modification to independent $Q$-learning in an attempt to stabilize the training process.

To deal with non-stationarity, agents need additional information to explain the changes in their environment. Existing approaches exploit either **(i)** multi-agent communication or **(ii)** centralized training of decentralized policies to provide the required additional information to agents. In communication based approaches (Sukhbaatar et al., 2016; Foerster et al., 2016; Lazaridou et al., 2017; Cao et al., 2018), agents use an emergent language to jointly execute a task. Such approaches require communication between agents even after the training is over. This needs additional computational resources and communication infrastructure which limits the applicability of these methods.

The centralized training and decentralized execution paradigm can be thought of as learning with the help of a coach that coordinates activities while training. Once the training process is over, agents can independently take decisions based on their local observations (Gupta et al., 2017). These approaches learn centralized critic(s) for all agents along with decentralized policies. While some approaches only support cooperative agents (Sunehag et al., 2018; Foerster et al., 2018b; Rashid et al., 2018; de Witt et al., 2018) others support competitive agents as well (He et al., 2016; Lowe et al., 2017; Iqbal & Sha, 2019). MA-SAC follows the centralized training and decentralized execution paradigm and it supports both cooperative and competitive agents.

MA-SAC is a maximum-entropy based algorithm. In the single-agent RL literature, maximum-entropy variant of many standard RL algorithms have been proposed like soft-$Q$-learning (Haarnoja et al., 2017) and soft-actor-critic (Haarnoja et al., 2018a). These algorithms are derived by casting the problem of learning optimal behavior as an inference problem in an appropriate graphical model (Levine, 2018). This not only provides a different perspective on the control problem, it leads to algorithms that are more stable and easier to train (Haarnoja et al., 2018a). A multi-agent variant of soft-$Q$-learning exists (Wei et al., 2018), but it is only applicable in cooperative environments. Note that MA-SAC is just one of the algorithms that can be derived using our proposed framework. As in the case of single-agent reinforcement learning, in the multi-agent setup as well, one can derive maximum entropy variants of other algorithms based on our proposed probabilistic model.

Most relevant to our work is the multi-agent variant of DDPG (Lillicrap et al., 2016) algorithm (called MADDPG) that has been proposed in Lowe et al. (2017). MADDPG is based on an actor-critic framework where each agent has its own centralized critic. In Section 4.3, we show that we recover an algorithm that is very similar to MADDPG by setting a tunable temperature parameter used in MA-SAC to zero. In Section 5, we show that MA-SAC outperforms MADDPG on several tasks.

If one ignores the attention mechanism proposed in Iqbal & Sha (2019), the resultant algorithm is equivalent to MA-SAC. While the main contribution in Iqbal & Sha (2019) is an improvement in scalability of MADDPG using an attention based mechanism in the critic, it is mostly heuristic based and a principled derivation of the algorithm is missing. On the other hand, we provide a unified probabilistic framework for multi-agent RL and use it to derive MA-SAC.

## 3 NOTATION AND PRELIMINARIES

A *Markov Decision Process* (MDP), specified by the tuple $(\mathbb{S}, \mathbb{A}, p, r, \gamma)$, models an environment with a single trainable agent (Sutton & Barto, 2018). Here, $\mathbb{S}$ is the set of states, $\mathbb{A}$ is the set of actions, $p : \mathbb{S} \times \mathbb{A} \to \Delta(\mathbb{S})$ specifies environment dynamics, $r : \mathbb{S} \times \mathbb{A} \to \mathbb{R}$ is the reward function and $\gamma \in [0, 1]$ is the discount factor. We use $\Delta(\mathbb{S})$ to denote the set of all probability distributions over set $\mathbb{S}$. A *Markov Game* (MG) generalizes MDPs to support multiple agents in the environment and is specified by the tuple $(\mathbb{S}, \{\mathbb{A}_i\}_{i=1}^n, p, \{r_i\}_{i=1}^n, \gamma)$ (Littman, 1994). Here, $n$ is the number of agents in the environment. Symbols $\mathbb{S}$ and $\gamma$ have the same meaning as before. $\mathbb{A}_i$ and $r_i : \mathbb{S} \times \mathbb{A}_1 \times \cdots \times \mathbb{A}_n \to \mathbb{R}$ respectively denote the action space and reward function of agent $i$. The transition probability function, $p : \mathbb{S} \times \mathbb{A}_1 \times \cdots \times \mathbb{A}_n \to \Delta(\mathbb{S})$, now considers the actions taken by all agents. The same holds for the reward functions $r_1, r_2, \ldots, r_n$. In practical cases, it is also common to assume that each agent can only observe a part of the environment. Let $\mathbb{O}_i$ denote the observation space of agent $i$, then, a function $f_i : \mathbb{S} \to \mathbb{O}_i$ computes the observation of agent $i$ from the environment state $\mathbb{S}$.

Note that both MDP and MG simply specify the evolution of environment in response to the behavior of agent(s), and as such, they do not encode any notion of optimal behavior. An agent formulates its own notion of optimality which it strives to achieve by modulating its policy $\boldsymbol{\pi} : \mathbb{S} \to \Delta(\mathbb{A})$ where, as before, $\Delta(\mathbb{A})$ is the set of all probability distributions over the set of actions $\mathbb{A}$. If there are multiple agents, we use $\boldsymbol{\pi}_i : \mathbb{S} \to \Delta(\mathbb{A}_i)$ to denote the policy followed by agent $i$. When the environment is partially observable, the policy of an agent can only take its local observation as input and hence $\boldsymbol{\pi}_i : \mathbb{O}_i \to \Delta(\mathbb{A}_i)$.

In single-agent RL, the agent often solves the following optimization problem (either exactly or approximately) to learn optimal behavior:

$$J(\boldsymbol{\pi}) = \max_{\boldsymbol{\pi}} \mathbb{E}_{\boldsymbol{\pi}} \Big[ \sum_{t=0}^{T} r(\mathbf{s}^{(t)}, \mathbf{a}^{(t)}) \Big]. \tag{1}$$

The agent acts in the environment only for a finite number of time steps $T$ (a finite horizon problem). The state and action at time $t$ are given by $\mathbf{s}^{(t)} \in \mathbb{S}$ and $\mathbf{a}^{(t)} \in \mathbb{A}$ respectively. The agent uses its stationary policy to choose an action $\mathbf{a}^{(t)} \sim \boldsymbol{\pi}(\cdot | \mathbf{s}^{(t)})$ at each time step. In MARL setting, different agents may have different reward functions. Moreover, because agents act simultaneously in the same environment, the optimal behavior of an agent now depends on the behavior of other agents. The notion of a *best response* policy of an agent becomes important in this setup. Each agent wants

to maximize its own expected sum of rewards but has to do so in the context of the policies being followed by other agents.

## 4  THE PROPOSED MODEL

### 4.1  MDP FOR EACH AGENT

We use probabilistic graphical models to formalize the idea of a best response strategy. Note that from the perspective of each agent, all the other agents are part of the environment. Thus, we model the world, as seen by each of the agents, as a separate MDP. The MDP for $i^{th}$ agent is depicted in Figure 1. For now, let us ignore the variables $o_i^{(t)}$ and all arrows connected to them for $t = 1, 2, \ldots, T$. We will use these variables while deriving MA-SAC in the next two sections. For agent $j \neq i$, let $\mathbf{a}_j^{(t)} \sim \boldsymbol{\pi}_j(\cdot \mid \mathbf{s}^{(t)})$ where $\boldsymbol{\pi}_j$ is the policy being followed by agent $j$. As these policies are part of the environment from the perspective of agent $i$, they have been encoded into the graphical model in Figure 1. In this MDP, the probability of transitioning from state $\mathbf{s}^{(t)}$ to state $\mathbf{s}^{(t+1)}$ is given by:

$$p(\mathbf{s}^{(t+1)} \mid \mathbf{s}^{(t)}, \mathbf{a}_i^{(t)}) = \int_{\mathbf{a}_{-i}^{(t)}} p(\mathbf{s}^{(t+1)} \mid \mathbf{s}^{(t)}, \mathbf{a}_i^{(t)}, \mathbf{a}_{-i}^{(t)}) \prod_{j \neq i} \boldsymbol{\pi}_j(\mathbf{a}_j^{(t)} \mid \mathbf{s}^{(t)}) \, d\mathbf{a}_{-i}^{(t)}. \tag{2}$$

Here, $\mathbf{a}_{-i}^{(t)}$ represents the actions taken by all agents except agent $i$ at time $t$. The first term inside integral in equation 2 is the transition function of underlying Markov game. The term on left hand side is the transition function of MDP for agent $i$. As other agents learn along with agent $i$, their policy $\boldsymbol{\pi}_j$ changes and hence, from the perspective of agent $i$, the environment dynamics are non-stationary. Thus, one cannot satisfactorily use standard single-agent RL algorithms to train each agent independently. In Fig 1, we have enclosed the actions of all agents $j \neq i$ and the environment state $\mathbf{s}^{(t)}$ in a box to indicate that all these variables are part of the environment from the perspective of agent $i$.

In addition to observing the current state $\mathbf{s}^{(t)}$, if agent $i$ is also allowed to observe the actions taken by all other agents $\mathbf{a}_j^{(t)}$, then the state transition dynamics can be specified by using a stationary transition function (the first term inside the integral in equation 2). This motivates centralized training of agents, where they are allowed to use this additional information so that the environment appears stationary to them, hence aiding in the learning process. Although the agents learn in a centralized fashion, we will see in the next two sections that the learned policies are completely decentralized, thereby avoiding any communication overhead during the execution of policies.

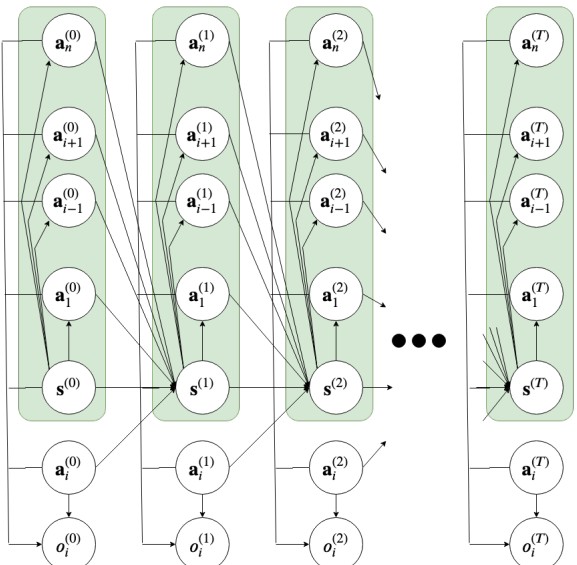

Figure 1: Augmented graphical model of MDP for agent $i$. Symbol $o_i^{(t)}$ denotes the optimality variable for agent $i$ at time $t$.

### 4.2  FINDING OPTIMAL POLICIES

As discussed earlier, MDPs themselves do not encode a notion of optimality. Hence, in Figure 1 we have augmented the graphical model of MDP for agent $i$ with additional optimality variables $o_i^{(t)}$, $t = 1, 2, \ldots, T$. Binary random variables $o_i^{(t)}$ indicate the optimality of action taken by agent $i$ at time $t$. Following (Levine, 2018), we assume that $r_i(\mathbf{s}^{(t)}, \mathbf{a}^{(t)}) \leq 0$ (this can be assumed without loss of generality for bounded reward

functions) and model the optimality variables as:

$$p(\mathrm{o}_i^{(t)} = 1 \mid \mathbf{s}^{(t)}, \mathbf{a}^{(t)}) = \exp\left(r_i(\mathbf{s}^{(t)}, \mathbf{a}^{(t)})\right). \tag{3}$$

Here $\mathbf{a}^{(t)} = [\mathbf{a}_1^{(t)}, \mathbf{a}_2^{(t)}, \ldots, \mathbf{a}_n^{(t)}]$ represents the joint action that has been taken by all agents. One way to find the optimal policy is by setting $\boldsymbol{\pi}_i^{(t)}(\mathbf{a}_i^{(t)} \mid \mathbf{s}^{(t)}, \mathbf{a}_{-i}^{(t)}) = p(\mathbf{a}_i^{(t)} \mid \mathbf{s}^{(t)}, \mathbf{a}_{-i}^{(t)}, \mathrm{o}_i^{(t:T)} = 1)$ (Levine, 2018). Note that it is enough to condition on $\mathrm{o}_i^{t'}$ for $t' = t, t+1, \ldots, T$ as the action at time $t$ is independent of $\mathrm{o}_i^{t'}$ for $t' < t$ given $\mathbf{s}^{(t)}$ and $\mathbf{a}_{-i}^{(t)}$. This policy is not stationary. Further, it relies on actions taken by other agents $\mathbf{a}_{-i}^{(t)}$ and hence is neither decentralized, nor practical to execute if all agents use a similar policy simultaneously. It is also known that such a policy is *risk seeking* in the sense that it always assumes the best case scenario and may choose to execute actions that will most likely lead to a bad outcome but may yield high reward infrequently due to unlikely state transitions (Levine, 2018). In real world, one has no control over environment dynamics and hence such risk seeking policies are undesirable.

To solve these issues, we use structured variational inference. We want to approximate the posterior distribution over trajectories $\boldsymbol{\tau} = \{\mathbf{s}^{(t)}, \mathbf{a}^{(t)}\}_{t=0}^T$ conditioned on all optimality variables for agent $i$ being one. However, we want the posterior to stay faithful to the environment dynamics to avoid risk seeking behavior. As the environment dynamics are fixed, optimal policies can be easily recovered from this posterior because of the structure of the approximating distribution (given in equation 4). We approximate the posterior over $\boldsymbol{\tau}$ using the following distribution:

$$q_{\boldsymbol{\theta}_i}(\{\mathbf{s}^{(t)}, \mathbf{a}^{(t)}\}_{t=0}^T) = p(\mathbf{s}^{(0)})\Big(\prod_{t=0}^{T}\prod_{j=1}^{n} \boldsymbol{\pi}_j(\mathbf{a}_j^{(t)} \mid \mathbf{s}^{(t)})\Big)\Big(\prod_{t=0}^{T-1} p(\mathbf{s}^{(t+1)} \mid \mathbf{s}^{(t)}, \mathbf{a}^{(t)})\Big). \tag{4}$$

We assume that the policy of agent $j$, i.e., $\boldsymbol{\pi}_j$, is parameterized by $\boldsymbol{\theta}_j \in \mathbb{R}^d$. To avoid notational clutter we suppress $\boldsymbol{\theta}_j$ and write $\boldsymbol{\pi}_j(\mathbf{a}_j^{(t)} \mid \mathbf{s}^{(t)})$, instead of $\boldsymbol{\pi}_j(\mathbf{a}_j^{(t)} \mid \mathbf{s}^{(t)}; \boldsymbol{\theta}_j)$.

There are two important features of this approximate posterior: **(i)** it uses the same model of environment dynamics as the underlying Markov game, and **(ii)** it assumes that agents take actions independently of each other. Because of **(i)**, the learned policies would not be risk seeking and, because of **(ii)**, they can be executed in a decentralized fashion. Also, note that while considering the MDP for agent $i$ given in Figure 1, $\boldsymbol{\theta}_i$ is the only parameter that can be adapted to bring the approximate posterior close to the true posterior. The parameters of all other policies are assumed to be constant.

Following the usual variational inference procedure (Blei et al., 2017), we write the expression for ELBO using the joint distribution $p(\boldsymbol{\tau}, \{\mathrm{o}_i^{(t)} = 1\}_{t=0}^T)$ and the approximate posterior distribution given in equation 4:

$$\mathrm{ELBO}_i(\boldsymbol{\theta}_i) = \mathbb{E}_{\boldsymbol{\tau}\sim q_{\boldsymbol{\theta}_i}}\big[\log p(\boldsymbol{\tau}, \{\mathrm{o}_i^{(t)} = 1\}_{t=0}^T) - \log q_{\boldsymbol{\theta}_i}(\boldsymbol{\tau})\big]$$

$$= \mathbb{E}_{\boldsymbol{\tau}\sim q_{\boldsymbol{\theta}_i}}\Big[\sum_{t=0}^{T}\log p(\mathbf{a}_i^{(t)}) + \sum_{t=0}^{T} r_i(\mathbf{s}^{(t)}, \mathbf{a}^{(t)}) - \sum_{t=0}^{T}\log \boldsymbol{\pi}_i(\mathbf{a}_i^{(t)} \mid \mathbf{s}^{(t)})\Big]$$

$$= \mathbb{E}_{\boldsymbol{\tau}\sim q_{\boldsymbol{\theta}_i}}\Big[\sum_{t=0}^{T} r_i(\mathbf{s}^{(t)}, \mathbf{a}^{(t)}) - \sum_{t=0}^{T}\log \boldsymbol{\pi}_i(\mathbf{a}_i^{(t)} \mid \mathbf{s}^{(t)})\Big] + \mathrm{const.} \tag{5}$$

We assume that the prior distribution over actions is uniform, hence $\log p(\mathbf{a}_i^{(t)})$ term has been absorbed in the constant in equation 5. Next, we derive MA-SAC that maximizes $\mathrm{ELBO}_i$ for all agents in an actor-critic framework.

### 4.3 MULTI-AGENT SOFT ACTOR-CRITIC

Although one can directly optimize $\mathrm{ELBO}_i$ over $\boldsymbol{\theta}_i$ for all agents $i$ using the REINFORCE gradient estimation trick (Williams, 1992), training agents in this way is hard due to high variance in the

gradient estimates. Thus, we cast the problem in an actor-critic framework. Define the function $Q_i(\mathbf{s}^{(t)}, \mathbf{a}^{(t)})$ as:

$$Q_i(\mathbf{s}^{(t)}, \mathbf{a}^{(t)}) = r_i(\mathbf{s}^{(t)}, \mathbf{a}^{(t)}) + \mathbb{E}_{\mathbf{s}^{(t+1)} \sim p}\Big[\mathbb{E}_{\mathbf{a}^{(t+1)} \sim \boldsymbol{\pi}}\Big[r_i(\mathbf{s}^{(t+1)}, \mathbf{a}^{(t+1)}) - $$
$$\log \boldsymbol{\pi}_i(\mathbf{a}_i^{(t+1)} \mid \mathbf{s}^{(t+1)}) + \mathbb{E}_{\mathbf{s}^{(t+2)} \sim p}\Big[\mathbb{E}_{\mathbf{a}^{(t+2)} \sim \boldsymbol{\pi}}\Big[\cdots\Big]\Big]\Big]\cdots\Big]. \qquad (6)$$

Here, $p$ is the transition function of Markov game and, with slight overloading of notation, we have assumed that $\boldsymbol{\pi}$ is a product distribution where each of the individual factors are given by $\boldsymbol{\pi}_j$, $j = 1, 2, \ldots, n$. One can recursively write $Q_i(\mathbf{s}^{(t)}, \mathbf{a}^{(t)})$ as:

$$Q_i(\mathbf{s}^{(t)}, \mathbf{a}^{(t)}) = r_i(\mathbf{s}^{(t)}, \mathbf{a}^{(t)}) + \mathbb{E}_{\mathbf{s}^{(t+1)} \sim p}\Big[\mathbb{E}_{\mathbf{a}^{(t+1)} \sim \boldsymbol{\pi}}\Big[Q_i(\mathbf{s}^{(t+1)}, \mathbf{a}^{(t+1)}) - \log \boldsymbol{\pi}_i(\mathbf{a}_i^{(t+1)} \mid \mathbf{s}^{(t+1)})\Big]\Big]. \qquad (7)$$

If the $\log \boldsymbol{\pi}_i$ term were not there in equation 7, $Q_i(\mathbf{s}^{(t)}, \mathbf{a}^{(t)})$ could have been interpreted as measuring the expected reward-to-go for agent $i$ when all agents start in state $\mathbf{s}^{(t)}$, take action $\mathbf{a}^{(t)}$ and thereafter follow the joint policy $\boldsymbol{\pi}$. This aligns with the commonly used notion of $Q$ function in RL. Note that we have a separate $Q$ function for each agent $i$. $Q_i$ uses actions of all agents at time $t$, i.e., $\mathbf{a}^{(t)}$ as input, as opposed to just using the action of agent $i$ because the actions of all agents are needed to sample the next state $\mathbf{s}^{(t+1)}$ from $p$.

The $\log \boldsymbol{\pi}_i$ term can be seen as an augmentation of the agent's reward function that increases the value of received rewards when the agent follows a policy with high entropy. We parameterize the function $Q_i$ using $\boldsymbol{\phi}_i \in \mathbb{R}^{d_Q}$. Using equation 7, $\boldsymbol{\phi}_i$ can be optimized by minimizing the following error:

$$\mathcal{E}_{Q_i}(\boldsymbol{\phi}_i) = \mathbb{E}_{(\mathbf{s}^{(t)}, \mathbf{a}^{(t)}) \sim \mathcal{D}}\Big[\big(Q_i(\mathbf{s}^{(t)}, \mathbf{a}^{(t)}) - \bar{Q}_i(\mathbf{s}^{(t)}, \mathbf{a}^{(t)})\big)^2\Big], \qquad (8)$$

where, $\mathcal{D}$ is the replay buffer. We use equation 7 to compute $Q_i(\mathbf{s}^{(t)}, \mathbf{a}^{(t)})$ using one-step lookahead. $\bar{Q}_i(\mathbf{s}^{(t)}, \mathbf{a}^{(t)})$ refers to the value of $Q_i$ under the current parameters $\boldsymbol{\phi}_i$. The lookahead term $Q_i(\mathbf{s}^{(t+1)}, \mathbf{a}^{(t+1)})$ appearing in equation 7 is also computed using the current value of parameters $\boldsymbol{\phi}_i$. Both $\bar{Q}_i(\mathbf{s}^{(t)}, \mathbf{a}^{(t)})$ and $Q_i(\mathbf{s}^{(t+1)}, \mathbf{a}^{(t+1)})$ are treated as constants while optimizing equation 8 over $\boldsymbol{\phi}_i$. In practice, all expectations are approximately evaluated by taking samples from the corresponding distributions. While we sample a fixed size batch of examples from $\mathcal{D}$, only one sample from $p$ and $\boldsymbol{\pi}$ is used to approximate equation 7. Also, note that we need a centralized training setup because computation of $\mathcal{E}_{Q_i}$ requires the ability to sample from policies of all agents.

This $Q_i$ estimate can now be used to find $\boldsymbol{\theta}_i$ that maximizes ELBO in equation 5. To optimize the policy of agent $i$ for choosing the correct action at time $t$, we have to maximize the term in nested expectation in equation 5 where action $\mathbf{a}_i^{(t)}$ is sampled. Thus, the objective (to be minimized) becomes:

$$\mathcal{E}_{\boldsymbol{\pi}_i}(\boldsymbol{\theta}_i) = -\mathbb{E}_{\mathbf{s}^{(t)} \sim \mathcal{D}}\Big[\mathbb{E}_{\mathbf{a}^{(t)} \sim \boldsymbol{\pi}}\Big[Q_i(\mathbf{s}^{(t)}, \mathbf{a}^{(t)}) - \log \boldsymbol{\pi}_i(\mathbf{a}_i^{(t)} \mid \mathbf{s}^{(t)})\Big]\Big]. \qquad (9)$$

We approximate the above expectation using samples. Gradients can either be computed using REINFORCE (Williams, 1992) or, if the policy permits, a reparameterization can be used to get lower variance gradient estimates. Algorithm 1 summarizes MA-SAC. In MADDPG, during optimization of equation 9, only actions of agent $i$ are sampled rather than sampling actions of all agents.

In practice, a temperature parameter $\alpha_i \in \mathbb{R}^+$ is multiplied to the $\log \boldsymbol{\pi}_i$ term in equation 7 and equation 9 to assign relative importance to entropy and reward maximization (Haarnoja et al., 2018a). Although there are methods for automatically selecting an approriate value of $\alpha_i$ (Haarnoja et al., 2018b), in all our experiments we treat it as a user specified hyperparameter.

Another important quantity that must be incorporated in equation 7 is the discount factor $\gamma$. One way to think about discounting in the probabilistic framework is by considering the possibility that an agent can die with probability $1 - \gamma$ at each time step and enter an absorbing state in which all subsequent rewards will be 0. The transition probability of going to a non-absorbing state now gets multiplied by $\gamma$ and hence $\gamma$ should be multiplied to the expectation term in equation 7 to incorporate discounted rewards. We do this in all our experiments.

We parameterize agent policies and $Q$ functions using neural networks. To improve learning stability, we also use target policy and $Q$ function networks. Additionally, each of our $Q$ function network consists of a pair of twin sub-networks and the minimum of the value produced by these sub-networks in a pair is taken as the output $Q$ value (Fujimoto et al., 2018).

## 5 EXPERIMENTS

### 5.1 ENVIRONMENTS

We use the multi-agent environments proposed in Lowe et al. (2017) to perform our experiments. These environments are built on the grounded communication framework that was proposed in Mordatch & Abbeel (2018). Each environment has a certain number of agents and landmarks. At each step, each agent observes a part of the global environment state and chooses an action from its discrete action space. In certain environments, the agents also have the ability to communicate with each other.

---
**Algorithm 1** Multi-agent Soft-Actor-Critic
---
Initialize parameters $\{\boldsymbol{\theta}_i, \bar{\boldsymbol{\theta}}_i, \boldsymbol{\phi}_i, \bar{\boldsymbol{\phi}}_i\}_{i=1}^n$
**for** each iteration **do**
    **for** each environment step **do**
        $\mathbf{a}_i^{(t)} \sim \boldsymbol{\pi}_i(\mathbf{a}_i^{(t)} \mid \mathbf{s}^{(t)})$, for $i = 1, 2, \ldots, n$
        $\mathbf{s}^{(t+1)} \sim p(\mathbf{s}^{(t+1)} \mid \mathbf{s}^{(t)}, \mathbf{a}^{(t)})$
        $\mathcal{D} \leftarrow \mathcal{D} \cup \{(\mathbf{s}^{(t)}, \{\mathbf{a}_i^{(t)}, r_i(\mathbf{s}^{(t)}, \mathbf{a}^{(t)})\}_{i=1}^n, \mathbf{s}^{(t+1)})\}$
    **end for**
    **for** each gradient step **do**
        $\boldsymbol{\theta}_i \leftarrow \boldsymbol{\theta}_i - \lambda_{\boldsymbol{\pi}} \nabla_{\boldsymbol{\theta}_i} \mathcal{E}_{\boldsymbol{\pi}_i}$, $i = 1, 2, \ldots, n$
        $\boldsymbol{\phi}_i \leftarrow \boldsymbol{\phi}_i - \lambda_Q \nabla_{\boldsymbol{\phi}_i} \mathcal{E}_{Q_i}$, $i = 1, 2, \ldots, n$
        $\bar{\boldsymbol{\theta}}_i \leftarrow \tau\boldsymbol{\theta_i} - (1 - \tau)\bar{\boldsymbol{\theta}}_i$, $i = 1, 2, \ldots, n$
        $\bar{\boldsymbol{\phi}}_i \leftarrow \tau\boldsymbol{\phi}_i - (1 - \tau)\bar{\boldsymbol{\phi}}_i$, $i = 1, 2, \ldots, n$
    **end for**
**end for**

---

In such cases, in addition to choosing a physical action, the agents also choose a communication action. The resultant communication message generated by an agent (represented as a one-hot encoded vector) is then broadcasted to all other agents where it becomes part of their observation in the next time step. Upon taking an action, each agent receives a reward from the environment which may potentially be distinct for different agents. Thus, based on the reward structure, the agents can be cooperative or competitive. We briefly describe the environments that we have used in our experiments here. For more details see Lowe et al. (2017).

**Cooperative navigation:** There are $n$ agents and $n$ landmarks in this environment. All agents get the same reward. This reward is calculated based on the distance of closest agent from each landmark. Additionally, agents occupy physical space and are penalized for colliding with each other. The optimal solution for agents is to spread out and *cover* all the landmarks simultaneously such that each landmark has an agent ideally overlapping with its position. Since everyone gets the same reward, this task requires cooperation among all agents. We use $n = 3$ in our experiments with this environment.

**Cooperative communication:** There are 3 landmarks, each colored differently. An agent (called listener) is rewarded based on its distance from the target landmark of a chosen color, however, it is unaware of this choice of color. A different immobile agent (called speaker) knows the color of target landmark. The speaker can only take communication actions and the listener can only take physical actions. The speaker receives the same reward as the listener and hence it must cooperatively communicate with the listener to transmit information about the color of target landmark.

**Predator-prey:** This environment mimics the classic predator-prey game where $k$ slower cooperating agents chase a faster adversary. All cooperating agents get a positive reward each time any of them collides with an adversary while the adversary gets a negative reward. This is a mixed cooperative-competitive environment where the $k$ cooperating agents compete with the adversary. We use $k = 3$.

**Physical deception:** There is one adversary and two cooperating agents in the environment. The environment also has two landmarks one of which is the target landmark. While the cooperating agents know which one is the target landmark, the adversary does not have this information. All agents want to reach the target landmark. The cooperating agents are rewarded based on minimum distance between target landmark and any of the cooperative agents. They are penalized based on distance between adversary and target landmark. Adversary gets rewarded for being close to the target landmark. The cooperative agents must go to both the landmarks so that the adversary is

unable to identify the target landmark based on the behavior of cooperative agents. This is again a mixed cooperative-competitive setting.

## 5.2 HYPERPARAMETERS AND OTHER IMPLEMENTATION DETAILS

We parameterize the $Q$ functions $Q_1, Q_2, \ldots, Q_n$ and agent policies $\boldsymbol{\pi}_1, \boldsymbol{\pi}_2, \ldots, \boldsymbol{\pi}_n$ using fully connected neural networks. All networks have two hidden layers having 128 units each. The hidden layers use ReLU activation function. We use Adam optimizer (Kingma & Ba, 2015) with learning rate 0.01. Following Lowe et al. (2017), we use Gumbel-Softmax (Jang et al., 2017) to approximate discrete sampling of actions in the policy networks (inverse temperature parameter of Gumbel-Softmax distribution was fixed to 1). We also use target policy and $Q$ networks. Target network parameters are exponentially moving averages of the corresponding policy or $Q$ network parameters. We update the target networks with each gradient step. For all experiments, we set $\tau = 0.01$ (in Algorithm 1) to compute the exponential average. For all environments, we train for 15000 episodes, each of length 25. One gradient step is taken after every 100 environment steps. We use a replay buffer of size $10^6$ which is populated in a round-robin fashion. A randomly sampled batch of 1024 examples from replay buffer is used for each gradient step. The value of $\alpha_i$ is empirically tuned separately for each environment. We use the following values: cooperative navigation: 0.02, cooperative communication: 0.005, predator-prey: 0.26 and physical deception: 0.01. These values were found using grid search. We use the same value of $\alpha_i$ for all cooperating agents in an environment. The value of discount $\gamma$ was set to 0.95 in all experiments.

## 5.3 COMPARISON WITH MADDPG

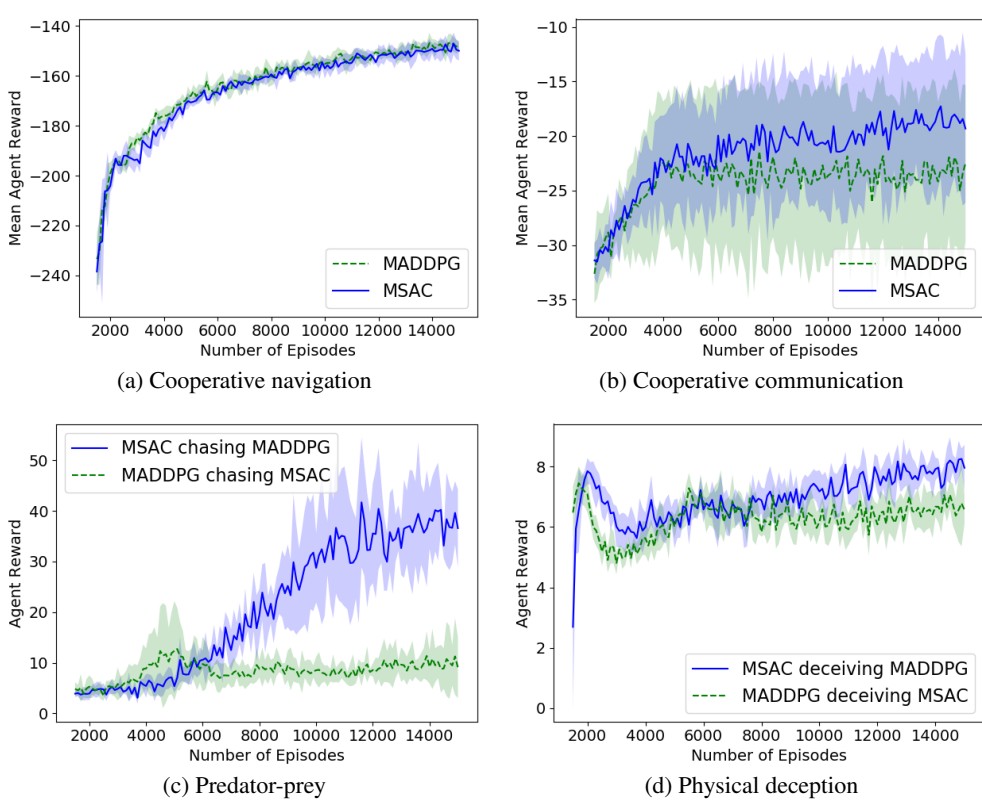

(a) Cooperative navigation

(b) Cooperative communication

(c) Predator-prey

(d) Physical deception

Figure 2: Comparison of MA-SAC (our approach) with MADDPG (Lowe et al., 2017). The temperature values used by MA-SAC controlled agents for each environment are given in Section 5.2. It can be seen that MA-SAC controlled agents outperform MADDPG controlled agents on majority of the tasks.

We chose to compare with the MADDPG algorithm (Lowe et al., 2017) for two reasons: **(i)** MA-SAC is closely related to MADDPG and a comparison with MADDPG allows us to demonstrate that the changes incorporated in MA-SAC do lead to better performance; and **(ii)** it was demonstrated in Lowe et al. (2017) that MADDPG outperforms state-of-the-art single agent reinforcement learning algorithms like DDPG (Lillicrap et al., 2016), TRPO (Schulman et al., 2015), DQN Mnih et al. (2015) and so on and thus it is a strong baseline.

Figure 2 shows the comparison between rewards obtained by MADDPG controlled agents and MA-SAC controlled agents on different environments. To obtain these plots, we executed the most recent learned policy at the time after every 100 episodes. The rewards were averaged over 100 independent episodes. Before testing, the policies were first converted into deterministic policies that select the most probable action. The entire process was independently repeated 5 times with different random seeds. The mean and standard deviation have been plotted in Figure 2.

For cooperative navigation and communication tasks, both agents were controlled by the same algorithm. For predator-prey and physical deception tasks, agents that compete with each other were controlled by different algorithms. For example, in Figure 2c, the red curve corresponds to a setting where the three cooperative agents are controlled by MA-SAC whereas the adversary is controlled by MADDPG. It can be seen that MA-SAC performs at least at par with MADDPG on all tasks and outperforms it on majority of the tasks.

### 5.4 OTHER REMARKS

The temperature parameter $\alpha_i$ plays an important role in determining the performance of MA-SAC. The $\log \boldsymbol{\pi}_i$ term in equation 9 augments the $Q$-function based on uncertainty of policy $\boldsymbol{\pi}_i$ in state $\mathbf{s}^{(t)}$. A very high value of $\alpha_i$ leads to policies that maximize entropy (i.e. behave randomly) and ignore the rewards obtained from the environment. At very low value of $\alpha_i$, the learned policies do not exhibit sufficient entropy and hence are susceptible to getting trapped in a local optima. Based on the scale of rewards obtained from the environment, the value of $\alpha_i$ must be tuned for achieving optimal performance. One can alternatively try to automatically tune $\alpha_i$ using techniques presented in (Haarnoja et al., 2018b), however, we leave these experiments for future work.

MA-SAC produces stochastic policies. Such policies are preferred during training as they are easier to train (Haarnoja et al., 2018a). Moreover, we believe that training with/against an agent that uses a stochastic policy would result in a more robust policy as it would have the same effect as training with an ensemble of deterministic policies as done in (Lowe et al., 2017). Once trained, it is often better to convert these stochastic policies to their deterministic counterparts to improve performance. This is in the same spirit as turning off $\epsilon$-greedy exploration during test time while performing $Q$-learning. As noted in Section 5.3, we follow this practice in our experiments.

Scaling efficiently as the number of agents increases is a common challenge associated with centralized training. In particular, the complexity of networks modeling $Q_i$ increases with the number of agents. A number of approaches have been proposed to parameterize the $Q$ network in various ways to improve scalability in certain settings (Yang et al., 2018; Rashid et al., 2018; Iqbal & Sha, 2019). For example, Yang et al. (2018) train each agent to play against the average opponent which allows training with hundreds of agents. These alternative parameterizations of $Q_i$ can naturally be integrated with MA-SAC. Finally, centralized training can still be done without access to policies of other agents as long as their actions are observable. To do so, Lowe et al. (2017) propose that each agent can train a local approximation to all opponents' policies in a supervised manner using their observed actions. This again can be used with MA-SAC. We leave detailed experiments in this direction for future work.

## 6 CONCLUSION

In this paper we posed the multi-agent RL problem as the problem of performing probabilistic inference in a graphical model where each agent views the environment as a separate MDP. We derived an off policy maximum entropy actor-critic algorithm based on the centralized training, decentralized execution paradigm using our proposed model. Our experimental results show that the proposed algorithm outperforms a strong baseline (MADDPG) on several cooperative and competitive tasks. As noted in Section 5.4, various existing ideas for parameterizing $Q$-functions (Yang et al., 2018; Rashid

et al., 2018; Iqbal & Sha, 2019) can be naturally integrated with MA-SAC to improve its scalability as the number of agents increases. Our framework can also be used for deriving maximum-entropy variants of other RL algorithms in the multi-agent setting. We leave these ideas for future work.

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
