# OpenReview forum: "Probabilistic View of Multi-agent Reinforcement Learning: A Unified Approach"
_ICLR.cc/2020/Conference — Reject_

### Official Review · AnonReviewer3 · 2019-10-23
**Official Blind Review #3**

**Rating:** 1

**Review:**

21st November Update: Thank you for your detailed response. I agree with the suggested future work and revisions to the paper. However, as no updates were made to the submitted paper, I will not be raising my score as the revisions represent significant changes that I cannot support acceptance of without further peer review. I encourage the authors to carefully consider all reviewers advice in updates to the paper for a future submission elsewhere, which I think will significantly improve the paper and its potential impact on the community.

--
This paper contributes a probabilistic framework for multi-agent RL and demonstrates a derivation of multi-agent SAC using it. The framework could be more broadly applicable if it included partial observability (as is often a requirement of multi-agent systems). The derivation could be improved by showing the full working for equation 5, as this would make the work self contained instead of assuming prior knowledge of ELBO by the reader.

The derived algorithm (previously published by Iqbal and Sha [ICML 2019] as noted in the paper) is then evaluated on 4 existing benchmark tasks against the baseline algorithm originally proposed with the environments - MADDPG. The environments represent a good range of multi-agent scenarios of suitable complexity to test modern deep RL algorithms. However, the empirical evaluation and methodology have issues that reduce the significance of their contribution.

On Page 8, it is noted that "the value of alpha_i is empirically tuned for each environment" but for which algorithm was it optimized? It is then noted that the values for alpha were "found using grid search" but details of the range of the search are not included nor details of how any other hyperparameters were set. Were other parameters tuned? If so please report all values searched for both algorithms and how they were set.

At the end of page 8, the caption of Figure 2 concludes "it can be seen that MA-SAC controlled agents outperform MADDPG controlled agents on majority of tasks." This statement is not supported by the graphs in this figure. I suspect Figures 2a and b show no significant difference as the confidence intervals overlap and that Figure 2d is not significantly different throughout training but may be with a small effect size at the current end of training. Figure 2d also looks like training for longer may be beneficial. Therefore, Figure 2c is the only environment that shows a significant improvement. Please provide further evidence that MA-SAC outperforms MADDPG or weaken this conclusion. It would also be interesting to investigate deeper, why MA-SAC shows such higher performance than MADDPG in the Predator-Prey domain.

On Page 9, the conclusion is reiterated and claimed to be in comparison to a state-of-the-art algorithm. However, the benefits of SAC over DDPG have been previously shown both in single agent [Haarnoja et al, ICML 2018] and multi agent domains [Iqbal and Sha, ICML 2019]. A stronger baseline to compare against would improve the significance of any resultant improvements.

The research direction is interesting but the earlier publication of the derived algorithm (Iqbal and Sha, ICML 2019) and the issues discussed above with the experimental results lead me to conclude that the contribution is not yet sufficient to warrant publication. With further work I believe this line of work could lead to a high impact publication, but feel the paper requires more changes than are feasible within the time frame of the ICLR rebuttal period.

Minor Comments:
- Page 4, "the transition function of underlying Markov game" -> the transition function of the underlying Markov game
- Page 9, "in Figure 2c, the red curve corresponds to" -> dark blue curve
- Page 9, "MA-SAC performs at least at par with MADDPG" -> at least on par with
- Page 9, "outperforms it on majority of the tasks" -> outperforms it on the majority of tasks

**Experience Assessment:**

I have published in this field for several years.

**Review Assessment: Checking Correctness Of Derivations And Theory:**

I carefully checked the derivations and theory.

**Review Assessment: Checking Correctness Of Experiments:**

I carefully checked the experiments.

**Review Assessment: Thoroughness In Paper Reading:**

I read the paper thoroughly.

---

> ### Author Response · Authors · 2019-11-11
> **Response to reviewer's comments**
>
> We thank the reviewer for their comments on the manuscript. These are truly valuable to us and we will surely incorporate the suggestions in the next version of the manuscript.
>
> REGARDING PARTIAL OBSERVABILITY
> From a practical point of view, one can simply train the policy networks of agents to use only the local observations made by them. The challenge is that complete state information is required for training the critics. A more careful derivation involving partial observability presents an interesting research problem and we will surely pursue it in future.
>
> REGARDING MISSING DETAILS
> We will add intermediate steps leading up to equation 5 in order to make the paper self contained as suggested by the reviewer. We will also add details about the process that we used for selecting various hyper-parameters to the supplementary material. The parameter alpha_i is relevant only for MA-SAC and hence it was optimized only for MA-SAC.
>
> REGARDING EXPERIMENTS
> We are in the process of executing more experiments, some with continuous action spaces, some with more complicated environments like StarCraft-II and so on. Also, all reviewers have pointed out some other experiments that may be added to the manuscript to strengthen it. We already have the results for environments with continuous action spaces and we will add these and many such experiments to the next version of the manuscript.
>
> We will also add experiments comparing our approach with Probabilistic Recursive Reasoning [1]. We believe that MA-SAC achieves significant performance improvement over MADDPG on predator-prey because of added stochasticity which is useful in competitive tasks. This information will be added to the paper.
>
>
> [1] Probabilistic Recursive Reasoning for Multi-Agent Reinforcement Learning. Y Wen, Y Yang, R Luo, J Wang, W Pan. ICLR 2019.

---

### Official Review · AnonReviewer2 · 2019-10-23
**Official Blind Review #2**

**Rating:** 3

**Review:**

Summary:
This paper proposes a new algorithm named Multi-Agent Soft Actor-Critic (MA-SAC) based on the off-policy maximum-entropy actor critic algorithm Soft Actor-Critic (SAC). Based on variational inference framework, the authors derive the objectives for multi-agent reinforcement learning. In experiments section, the authors compare the proposed algorithm with the previous algorithm called Multi-Agent Deep Deterministic Policy Gradient (MADDPG) on several multi-agent domain.

Comments:
- Based on inference, the authors derive the objectives as the equation (8) and (9). However, the proposed objectives are almost identical to SAC. First, the objectives for Q functions are just replacing \hat{Q} in the equation (7) in SAC by \bar{Q}, which has a very similar meaning. Also, the objectives for policy \pi are exactly the same as in the SAC with only the added index. Thus, the proposed algorithm seems to be a naive extension of SAC into multi-agent cases. To avoid such questions, authors need to emphasize the difference from simple extension.
- Is there a reason not to use additional neural networks to estimate value function like SAC even the proposed algorithm is based on SAC?
- Additional experimental results are needed to ensure the algorithm since there is no theoretical guarantee.

**Experience Assessment:**

I have read many papers in this area.

**Review Assessment: Checking Correctness Of Derivations And Theory:**

I assessed the sensibility of the derivations and theory.

**Review Assessment: Checking Correctness Of Experiments:**

I assessed the sensibility of the experiments.

**Review Assessment: Thoroughness In Paper Reading:**

I made a quick assessment of this paper.

---

> ### Author Response · Authors · 2019-11-11
> **Response to reviewer's comments**
>
> We thank the reviewer for their valuable suggestions. We will modify the manuscript to incorporate the proposed changes.
>
> REGARDING NOVELTY
> Our major contribution lies in providing the multi-MDP view of the environment where we model the environment as seen by each agent using a separate but related MDP. Using this model, we show that the multi-agent variant of soft actor-critic can be derived by applying simple techniques. As we have noted in the paper, even though we only derive an actor-critic based algorithm, one can also use the model that we have described in the paper to derive multi-agent variants of other algorithms like Q-learning. MA-SAC is simply a representative example.
>
> REGARDING USAGE OF VALUE FUNCTION
> We did experiment with the setting where value function is also used as done in the original soft actor-critic paper. However, on the tasks that we experimented with, we did not see any noticeable change in the performance of the algorithm or the training characteristics. Thus, in order to reduce the number of parameters being trained, we did not use the value function in our final implementation.
>
> REGARDING EXPERIMENTAL EVALUATION
> We are in the process of executing more experiments, some with continuous action spaces, some with more complicated environments like StarCraft-II and so on. Also, all reviewers have pointed out some other experiments that may be added to the manuscript to strengthen it. We already have the results for environments with continuous action spaces and we will add these and many such experiments to the next version of the manuscript.

---

### Official Review · AnonReviewer1 · 2019-10-24
**Official Blind Review #1**

**Rating:** 3

**Review:**

The paper extends soft actor-critic (SAC) to Markov games, or in other words multi-agent reinforcement learning setting. The paper is very nicely written, derives MA-SAC in a fairly general way, and introduces a variational approximation of the distribution over optimal trajectories which enables centralized training and decentralized execution. While I like the paper, I find the novelty aspect of it quite limited, since it's quite a straightforward combination of centralized training and decentralized execution idea with an algebraic extension of SAC to Markov games. The paper would have been much stronger if it had a much more thorough evaluation of the properties and limitations of MA-SAC as well as better comparison with the related work.


Questions/comments:

1. One of the key points of the paper is equation 4 that proposes the variational approximation of the distribution over trajectories. The authors assume that agents take actions independently which enables decentralized execution. However, it looks like this construction neglects the fact that optimal policies *must* take into account the other agents. It seems that with q structured this way, dependencies between agent actions are not taken into account even when training is centralized (all equations 5-7 fully factorize, neglecting all dependencies). In other words, given the proposed q, what is the benefit of centralized training?

2. Following up on the previous question, from Levine (2018) we know that Eq. (3) results in a particular soft-Q function that can be computed using the forward-backward algorithm (assuming the knowledge of the dynamics), which would account for dependencies between agent policies/actions. On the other hand, it's unclear whether/how the Q function obtained through centralized training (Eq. 8) approximates the optimal soft-Q function. Can the authors comment on that?

3. As mentioned, the proposed MA-SAC is really a fairly straightforward extension of SAC to Markov games. What could make the paper interesting in my opinion is a much more detailed (experimental) analysis of the approximations the authors had to make in order to enable decentralized execution and the corresponding advantages and limitations. The current evaluation falls short on that front as it just shows that the proposed algorithm works better than MADDPG in a few standard multi-agent environments.

4. Although the authors position this paper as the first that introduces a probabilistic perspective on RL in multi-agent systems, there is other recent work (https://arxiv.org/abs/1901.09207) that already does that and, in fact, takes one more step and enables decentralized training with (probabilistic) reasoning about other agents. Discussion of advantages/disadvantages and comparison with the previous work I see as necessary.

----

I acknowledge that I have read the author's response. My assessment of the paper stays the same.

**Experience Assessment:**

I have published one or two papers in this area.

**Review Assessment: Checking Correctness Of Derivations And Theory:**

I carefully checked the derivations and theory.

**Review Assessment: Checking Correctness Of Experiments:**

I carefully checked the experiments.

**Review Assessment: Thoroughness In Paper Reading:**

I read the paper thoroughly.

---

> ### Author Response · Authors · 2019-11-11
> **Response to reviewer's comments**
>
> We thank the reviewer for their valuable feedback. These suggestions would surely help us in improving the quality of the manuscript.
>
> HOW CAN AGENTS COORDINATE DESPITE THE INDEPENDENCE ASSUMPTION IN EQUATION 4?
> The policy for each agent i is trained by optimizing equation 9. Note that equation 9 involves computation of Q_i which in turn requires sampling actions of all agents given the current environment state. Assume for a moment that the policies being followed by all agents except agent i were fixed. In this case, agent i will tend to choose an action that is the best response to the policies being followed by other agents. Thus, because of the use of centralized training, the agent is implicitly learning to consider the possible actions that can be taken by other agents and act accordingly. Conditioning on actions of other agents and then hallucinating their actions during testing (as done in [1]) is useful when agents are trained in a decentralized fashion. We will further clarify the argument and add a comparison with [1] in the next version of the manuscript.
>
> REGARDING EXPERIMENTAL EVALUATION
> We are in the process of executing more experiments, some with continuous action spaces, some with more complicated environments like StarCraft-II and so on. The reviewer has rightly pointed out that exploring different properties of MA-SAC, understanding the role played by different assumptions and comparing with stronger baselines would further strengthen the paper. We already have the results for environments with continuous action spaces and we will add these and many such experiments to the next version of the manuscript.
>
> REGARDING NOVELTY
> Our major contribution lies in providing the multi-MDP view of the environment where we model the environment as seen by each agent using a separate but related MDP. Using this model, we show that the multi-agent variant of soft actor-critic can be derived by applying simple techniques. As we have noted in the paper, even though we only derive an actor-critic based algorithm, one can also use the model that we have described in the paper to derive multi-agent variants of other algorithms like Q-learning. MA-SAC is simply a representative example.
>
> REGARDING DERIVATION OF ACTUAL SOFT Q-FUNCTION ALONG THE LINES OF LEVINE (2018)
> We have a derivation for computing the analogous soft Q-function for multi-agent setting using the forward-backward algorithm as done in Levine (2018). We will add these details in the supplementary material.
>
>
> [1] Probabilistic Recursive Reasoning for Multi-Agent Reinforcement Learning. Y Wen, Y Yang, R Luo, J Wang, W Pan. ICLR 2019.

---

### Public Comment · ~Yaodong_Yang1 · 2019-09-26
**multi-agent soft-actor-critic has been developed by multiple previous work, do PLEASE cite them**

Hello:

Thanks for presenting this work.

However, we have serious concerns about the novelty of this work. The effort of the mapping the multi-agent learning question into the probabilistic inference on the graphical model, i.e. the multi-agent soft learning, has actually been done by multiple previous work, however, the author has cited none of them.

1. Probabilistic Recursive Reasoning for Multi-Agent Reinforcement Learning
    Y Wen, Y Yang, R Luo, J Wang, W Pan
    ICLR 2019

2. A Regularized Opponent Model with Maximum Entropy Objective
    Z Tian, Y Wen, Z Gong, F Punakkath, S Zou, J Wang
    IJCAI 2019

3. Wei, Ermo, et al. "Multiagent soft q-learning." 2018 AAAI Spring Symposium Series. 2018.

4. Balancing two-player stochastic games with soft q-learning
    J Grau-Moya, F Leibfried, H Bou-Ammar
    AAAI 2018

More importantly, the author claim this work to be "a unified view", however, it turns out if you define your optimality variable in the graphical model solely based on mapping the single-agent case to the multiagent case, i.e, P(o=1 | s1, a1, a2) \propotional exp(R(s,a1,a2)), then this framework could NOT even solve the simple zero-sum setting in multi-agent learning.


END.

---

> ### Author Response · Authors · 2019-10-02
> **Thanks for pointing out interesting related works**
>
> Thank you for pointing out these related works. We have gone through these papers carefully and we will cite them in our paper.
>
> We will revise the manuscript to:
>
> 1. Add the following paragraph to the Related Works section:
> "[1] integrates probabilistic recursive reasoning while training several maximum entropy RL agents in a decentralized fashion. [2] proposes a probabilistic model where, conditioned on the optimality of other agents, each cooperative agent aims at maximizing its own probability of being optimal. In [4], the objective is to train a sub-optimal policy for agents in two-player video games. This policy must be close to a reference policy in Kullback-Leibler divergence. While [2] and [4] are not general enough to be used across all our experiments, we compare MA-SAC with [1] in Section 5."
>
> 2. Add experiments comparing MA-SAC with PR2 on both cooperative and competitive tasks
>
> Our comments on these related works are as follows.
>
> We would like to emphasize that the way we model each agent using a separate but related MDP is one of our major contributions. It provides a different perspective on the problem and, as we have also noted in our paper, it allows one to derive a variety of efficient MARL algorithms of which MA-SAC is an example.
>
> In [1], it has been shown that PR2 does not perform well against MADDPG on competitive tasks from the multiagent particle environment while our experiments show that MA-SAC outperforms MADDPG. We will add comparison against PR2 on multiple cooperative and competitive tasks to the revised version of our manuscript. Note that even MA-SAC can be potentially trained in a decentralized fashion by using an opponent modelling trick in the same way as it was used in MADDPG. We will explore this issue in our experiments.
>
> Approaches [2] and [3] are restricted to cooperative settings only whereas our formulation allows cooperative as well as competitive agents. Approach [3] has already been cited in the paper.
>
> Approach [4] has been developed for two agents whereas our framework supports more than two agents as well.
>
> We will only add experiments with PR2 because approaches proposed in [2-4] are either not suitable for competitive scenarios or do not support an arbitrary number of agents.
>
> When we say our proposed framework presents a unified view, what we mean is that many different algorithms (Q-learning based, policy gradient based and actor-critic based) can be derived using our framework.
>
> Thank you!
>
> [1] Probabilistic Recursive Reasoning for Multi-Agent Reinforcement Learning. Y Wen, Y Yang, R Luo, J Wang, W Pan. ICLR 2019.
>
> [2] A Regularized Opponent Model with Maximum Entropy Objective. Z Tian, Y Wen, Z Gong, F Punakkath, S Zou, J Wang. IJCAI 2019.
>
> [3] Multiagent soft q-learning. Wei et al. AAAI Spring Symposium Series 2018.
>
> [4] Balancing two-player stochastic games with soft q-learning. J Grau-Moya, F Leibfried, H Bou-Ammar. AAAI 2018.

---

### Decision · Program_Chairs · 2019-12-19

**Decision:**

Reject

**Comment:**

The paper takes the perspective of "reinforcement learning as inference", extends it to the multi-agent setting and derives a multi-agent RL algorithm that extends Soft Actor Critic. Several reviewer questions were addressed in the rebuttal phase, including key design choices. A common concern was the limited empirical comparison, including comparisons to existing approaches.